# Microstructure, Interface and Strengthening Mechanism of Ni-CNTs/AZ91 Magnesium Matrix Composites

**DOI:** 10.3390/ma15227946

**Published:** 2022-11-10

**Authors:** Zhengzheng Liu, Shaoyong Qin, Wuxiao Wang, Jian Liu, Dongchao Liu, Xiaogang Chen, Wuzhao Li, Bingchu Mei

**Affiliations:** 1School of Material Science and Engineering, Xi’an University of Technology, Xi’an 710048, China; 2Faculty of Printing, Packaging Engineering and Digital Media Technology, Xi’an University of Technology, Xi’an 710054, China; 3State Key Laboratory of Advanced Technology for Materials Synthesis and Processing, Wuhan University of Technology, Wuhan 430070, China

**Keywords:** Ni-CNT, magnesium matrix composites, strengthening mechanism, ultrasonic treatment, semi-solid stirring

## Abstract

Ni-CNTs/AZ91 magnesium matrix composites were fabricated by ultrasound treatment combined with a semi-solid stirred method for the first time. The agglomerated spherical Ni-CNTs transferred from spherical shape to clear tubular shape after pre-dispersion treatment. For the Ni-CNTs/AZ91 magnesium matrix composite prepared by semi-solid stirring followed by ultrasonic treatment, Ni-CNTs were evenly distributed in the magnesium matrix or wrapped on the β (Mg_17_Al_12_) phase. Mg_2_Ni were formed at the interface of the magnesium matrix and CNTs by in-situ reaction, which significantly improved the interface bonding strength of CNTs and the Mg matrix. The tensile strength and elongation of 1.0wt.% Ni-CNTs/AZ91 magnesium matrix composites were improved by 36% and 86%, respectively, compared with those of AZ91 matrix alloy. After Ni-CNTs were added to AZ91 matrix alloy, more dimples were observed at the fracture surface. The fracture behavior of Ni-CNTs/AZ91 composite was transformed from a cleavage fracture of AZ91 matrix alloy to a quasi-cleavage fracture. Meanwhile, the CNTs dispersed near the fracture showed a “pull-out” state, which would effectively bear and transfer loads. The strengthening mechanism of CNTs was also discussed.

## 1. Introduction

Magnesium alloy possesses various advantages, such as good machinability, high specific strength, and high specific stiffness [1,2,3,4,5,6], leading to a promising application in the fields of medical devices, new energy vehicles, aerospace, and electronic devices [7,8,9,10,11,12]. However, the poor mechanical properties of magnesium alloy, low strength, low elastic modulus, and poor plasticity, highly limit its practical application. It is an effective approach to improving the mechanical properties of magnesium alloy, which have attracted much attention in recent years. Adding SiC [13], TiC [14], Al_2_O_3_ [15], CNT [16] to magnesium alloy effectively enhanced its mechanical properties. CNT was first discovered by Iijima in 1991 [17,18]. It has excellent mechanical properties with a tensile strength of 50–200 GPa and elastic modulus of 1 TPa, and is widely considered a promising reinforcement [19,20,21,22]. Hayder et al. [23] prepared multi-walled carbon nanotubes (MWCNTs)/Mg matrix composite material by powder metallurgy. The research showed that the addition of CNTs in the magnesium matrix had a certain improvement in the compressive strength of the composite. With the increase of the volume fraction of the carbon nanotube reinforcement, the wear resistance was gradually improved. However, CNTs’ advantages could not be fully exploited since the ball milling process destroyed their structure. Uozumi et al. [24] prepared MWCNTs/Mg matrix composite material by the squeeze casting method. The integrity of the CNTs was good, but the wettability between matrix and reinforcement was poor, resulting in adverse interface bonding. In order to solve the wettability of CNTs and Mg, Han et al. [25] prepared Ni-CNTs/Mg matrix composite material by hot extrusion. Mg_2_Ni was generated by in situ reaction at the interface between the CNTs and the Mg matrix. There is an effective interfacial binding between the CNTs and the Mg matrix, but the dispersion of CNTs in the Mg matrix is poor. Without the advantage of a high length-diameter ratio, CNTs cannot effectively improve the strength and toughness of the magnesium matrix. AZ91 magnesium alloy has good castability, but poor strength and toughness. The strength and toughness of the composites were significantly improved by adding Ni-CNTs to the AZ91 magnesium matrix. There are no relevant studies on the interface, microstructure and mechanical properties of Ni-CNTs to strengthen AZ91 magnesium matrix composites at present.

In this paper, ultrasound treatment combined with a semi-solid stirred method was used to prepare the Ni-CNTs-reinforced AZ91 magnesium matrix composites for the first time. The agglomeration of Ni-CNTs was dispersed by ultrasonic crushing in the pre-dispersion process. Ni-CNTs, which aggregated in the AZ91 magnesium matrix were dispersed by melt ultrasound in the smelting process. The dispersion of Ni-CNTs, the dispersion of Ni-CNTs in the matrix material, the interface between Ni-CNTs and the matrix material and the mechanism of enhancing mechanical properties were systematically studied.

## 2. Experimental Materials and Methods

The primary experimental materials were presented in Table 1. Ni-CNTs and TNADIS dispersant were obtained from Chengdu times nano. The preparation process of Ni-CNTs/AZ91 magnesium matrix composite is shown in Figure 1. Chemical dispersants were added into anhydrous ethanol. Then, Ni-CNTs were fed into the solution to obtain suspension of the Ni-CNTs, which were ultrasonically dispersed for 2 h with an ultrasonic power of 550 W. After that, magnesium powder was added to the suspension of CNTs, and magnetic stirring was carried out for 3 h. After drying at 50 °C, Ni-CNTs-Mg powder mixed powder was obtained. Finally, the mixed powder was placed in a mold and pressed into blocks by a press at a pressure of 780 MPa to obtain Ni-CNTs/Mg pre-dispersed blocks (diameter: 2.5 mm, height: 1.8 mm), which retained the dispersed state of Ni-CNTs on the surface of the magnesium powder particles to prevent their re-agglomeration. After the powder mixture is directly added to the alloy liquid in the smelting process, the powder will be suspended on the surface of the alloy liquid because of its light weight, which is not conducive to the dispersion of carbon nanotubes in the matrix, and the dispersed blocks will avoid this situation to a large extent. Then, the dispersed block was added to the magnesium alloy liquid and subjected to ultrasound-assisted semi-solid agitation, and the Ni-CNTs/AZ91 magnesium matrix composite was cast. In this paper, the AZ91 alloy is melted by well resistance furnace (model: SG2-7.5-12A), and the melting process is shown in Figure 2. The AZ91 alloy ingot was completely melted at 720 °C. Then, the melting furnace temperature was lowered to a semi-solid temperature of 630 °C, compacted Ni-CNTs/Mg blocks were added, and semi-solid stirring was performed at 630 °C and 500 r/min for 5 min. After that, the mixed melt was heated to 700 °C, the ultrasonic vibrator was inserted into the melting furnace, preheated for 30 min, and ultrasonic treatment was carried out with a power of 1 KW for 10 min. Subsequently, the resistance furnace was heated to 720 °C and held for 5 min. Eventually, the composite melt was poured into the mold and solidified to obtain Ni-CNTs/AZ91 magnesium matrix composites. To prevent the oxidation combustion of the alloy during the melting process, a little RJ-2 covering agent was sprinkled on the magnesium alloy metal block and the whole smelting process was carried out under an argon atmosphere. The AZ91 alloy ingots were prepared by the same preparation process for comparison.

Phase analysis of Ni-CNTs/AZ91 composites was carried out by X-ray diffractometer (model: XRD-7000), the wavelength was 0.154 nm, the anode material was Cu, the 2θ range of measurement was 20–90°, the step size was 0.02°, and the scanning speed was 4°/min. Metallographic samples were prepared by a standard sample preparation process, then the samples were etched with a 5% nitric acid ethanol solution. The microstructure of the samples was characterized by a Zeiss scanning electron microscope (model: Merlin Compact) and transmission electron microscope (model: Tecnai G2 F30 S-Twin). The acceleration voltage of SEM was 10 KV, and all SEM images were secondary electron images. The acceleration voltage of TEM was 300 KV. The composite tensile specimens with different content of CNTs were machined by electrical discharge machining and polished with sandpaper, and the dimension of the specimen was as shown in Figure 3. The HT-2040 universal tensile testing machine was used to carry out the tensile test at room temperature, and the tensile speed was set at 0.5 mm/min. At least three tests were repeated for each type of material.

## 3. Results and Discussion

### 3.1. Microstructural Analysis

Figure 4 shows the morphology of Ni-CNTs before and after pre-dispersion. It can be seen that Ni-CNTs are agglomerated with large particle sizes due to the strong van der Waals force between Ni-CNTs before pre-dispersion [26], hence it is difficult to take full advantage of the high aspect ratio as super reinforcement. The morphology of Ni-CNTs obtained after pre-dispersion treatment is shown in Figure 4b. The agglomerated spherical Ni-CNTs transferred from spherical shape to clear tubular shape after pre-dispersion treatment. There are two reasons for this: first, with the addition of TNADIS dispersant to Ni-CNTs, the lyophilic and lipophilic groups of the dispersant are attached to the nanotubes to weaken the van der Waals forces between the nanotubes; second, the cavitation effect was generated in the suspension of Ni-CNTs by ultrasonic treatment, the shock waves formed by microbubble breakage destroy the agglomeration of Ni-CNTs.

The phase composition of Ni-CNTs/AZ91 magnesium matrix composite was characterized by X-ray diffraction analysis, as shown in Figure 5. It can be seen from the X-ray diffraction pattern that α-Mg and β-Mg_17_Al_12_ phases are the main phases in the composite material, the diffraction peak of C (CNTs) is found at 26° and 42°, the diffraction peaks of Mg_2_Ni are found at 40°, 44° and 52°. After adding a small amount of Ni-CNTs to the AZ91 magnesium matrix, the interface reactant Mg_2_Ni was generated by in-situ reaction at the interface between matrix and CNTs, which improved the interface binding ability between the matrix and CNTs, gave full play to the advantage of the ratio of height to aspect of CNTs, and achieved the purpose of improving the strength and toughness of the magnesium alloy. Ni is pre-plated on the surface of CNTs, so no Mg_2_Ni dispersed separately in the magnesium matrix or around the CNTs.

To explore the distribution of CNTs and Mg_2_Ni in the composite, the microstructure of the composite was observed by SEM and EDS analysis, as shown in Figure 6. After energy spectrum analysis, some regions (labeled as ABCD) in Figure 6a,b are respectively displayed in Figure 6c–f. It can be known that region A in Figure 6a is CNTs, region B is the Magnesium matrix, region C in Figure 6b is the CNTs, region D is β-Mg_17_Al_12_. It can be found that the morphology of the CNTs is a clear tubular structure, there are two main distribution modes in the composite material: one contains a large number of CNTs, evenly distributed in the Mg alloy matrix; the other contains a small number of winding distributions of CNTs on the β-Mg_17_Al_12_ phase. This phenomenon can be explained as follows. During semi-solid stirring, a small amount of solid phase α-Mg exists, and strong collisions occurred between the solid particles and CNTs, which contributes to the dispersion of CNTs in the matrix alloy. During the ultrasonic treatment, the supersonic cavitation effect was generated in the melt. The breakup of tiny bubbles produces shock waves that break up CNTs gathered in the alloy solution to achieve the secondary dispersion of CNTs in the matrix alloy. This indicates that ultrasound treatment combined with a semi-solid stirred method can effectively prevent Ni-CNTs from agglomeration in the fabrication of the Ni-CNTs/AZ91 magnesium matrix composites.

### 3.2. Interface Analysis

To analyze the interface bonding mode between CNTs and the Mg matrix, TEM analysis was carried out on the composites, and the distribution maps of CNTs, Mg, and Mg_2_Ni in Ni-CNTs/AZ91 composites were obtained. Figure 7 presents CNTs in Ni-AZ91/CNTs composite materials, Mg and Mg_2_Ni distribution. By scanning the A region at the interface in Figure 7a with TEM energy spectrum, it can be found that Mg, C, and Ni are distributed at the interface between the CNTs and the Mg matrix. As shown in Figure 7b–d, the results of the surface scan area are consistent with the X-ray diffraction pattern, which can better explain the existence of the interface reactant Mg_2_Ni.

Figure 8 shows the electron diffraction pattern and the composite interface structure. To explore the interface bonding relationship among CNTs, Mg, and Mg_2_Ni, selected electron diffraction was carried out in area B at the interface of Figure 8a, and the selected diffraction pattern was calibrated. It was found that Mg_2_Ni has three different crystal plane spacings at the interface, which correspond to (200)_Mg2Ni_, 0.224 nm, and (107)_Mg2Ni_, 0.147 nm, (215)_Mg2Ni_, 0.141 nm, respectively. In addition, two kinds of C phases with different crystal plane spacing can be found, which are 0.357 nm, 0.206 nm respectively, corresponding to (004)_C_, and (114)_C_ crystal planes. This is consistent with the results obtained by X-ray diffraction analysis.

To deeply understand the interface information of CNTs, Mg, and Mg_2_Ni, area A of the interface in Figure 7a was further enlarged, as shown in Figure 8b. The crystal morphologies are (004)_C_, (200)_Mg2Ni_, and (101)_Mg_, and the corresponding crystal plane spacing is 0.357 nm, 0.225 nm, and 0.250 nm, respectively. The existence of the interface layer Mg_2_Ni between the CNTs and the magnesium matrix can be exhibited. Meanwhile, there is a coherent interface relationship between Mg_2_Ni and the magnesium matrix by further analysis of the crystal structure, and the coherent angle between the two planes is 139.6°. Figure 8c is a schematic diagram of the coherent interface. The existence of the interface coherent relationship between Mg_2_Ni and the magnesium matrix will enhance the connection between the magnesium matrix and the CNTs, and improve the interface bonding strength between the CNTs and the matrix materials. Under the connection of interface reactant Mg_2_Ni, the magnesium matrix can form a strong interface bonding with CNTs, which can effectively enhance the strength and toughness of composite materials [26].

### 3.3. Mechanical Properties

Figure 9 shows the tensile test results of AZ91 alloy and 1.0 wt.% Ni-CNTs/AZ91 magnesium matrix composites at room temperature. Figure 10 shows the tensile stress-strain curves of the two materials. The tensile strength of the AZ91 alloy and the Ni-CNTs/AZ91 magnesium matrix composite was 140 MPa and 190 MPa, the elongation was 4.2% and 7.8%, respectively. It can be observed that the tensile strength and elongation of the composite materials after adding Ni-CNTs to the AZ91 alloy were superior to the AZ91 alloy. Compared with the AZ91 matrix alloy, the tensile strength and elongation of the 1.0 wt.% Ni-CNTs/AZ91 magnesium matrix composite were enhanced by 36% and 86%, respectively. The change in the mechanical properties can be explained as follows: firstly, during the tensile process of the composites, the tensile load is transferred from the matrix to the CNTs. With the gradual increase of the tensile load, the CNTs change from the bending state to the straightening state, until the final fracture, and the CNTs bear the main load. Secondly, the interface reactants formed between the CNTs and the Mg matrix play a role in connecting the CNTs and the Mg matrix, which improves the interface bonding strength.

To analyze the strengthening mechanism of Ni-CNTs/AZ91 magnesium matrix composite, the fracture morphology of two materials (AZ91 matrix material and 1.0 wt.% Ni-CNTs/AZ91 magnesium matrix composite) was observed by SEM, as shown in Figure 11. The Mg alloy has a dense hexagonal crystal structure, which can only be plastically deformed by basal plane slip and twin, generally. Under the action of external forces, magnesium alloy grain deformation coordination is insufficient, and it is difficult to deform at room temperature. Figure 11b shows the fracture morphology of 1.0 wt.% Ni-CNTs/AZ91 magnesium matrix composites. Tearing edges and many dimples can be found in the fracture surface, the fracture surface is rough with large surface undulation, and the section is a quasi-cleavage fracture mixed with cleavage plane and dimple. The “pull-out” state of a small number of CNTs can be seen from the fracture morphology of the high magnification region in Figure 11c. The drawing process of the composite material is shown in Figure 12. It can be seen from Figure 4b that the pre-dispersed Ni-CNTs are in a state of bending, while the Ni-CNTs found at the tensile fracture are in a straight state, which indicates that Ni-CNTs go through four stages of bending, stretching, bridging, and fracture in the process of stretching [27]. This process can be visually demonstrated more in Figure 12. This indicates that the interface reactant Mg_2_Ni generated by the in-situ reaction of Ni-CNTs added to the AZ91 matrix alloy can effectively improve the binding strength of the matrix materials and CNTs. The CNTs dispersed near the fracture can effectively carry and transfer the load, inhibit the fracture of the matrix, and help to enhance the strength and plasticity of the composite.

## 4. Conclusions


Ni-CNTs were tubular with obvious slenderness ratio advantage after the pre-dispersed treatment of Ni-CNTs. Meanwhile, CNTs in Ni-CNTs/AZ91 magnesium matrix composites prepared by ultrasonic-assisted treatment combined with semi-solid stirring casting exist in two states. Ni-CNTs were evenly distributed in the magnesium matrix, or wrapped on the β phase. Mg_2_Ni was generated by in-situ reaction at the interface of magnesium matrix and CNTs, which effectively enhanced the interfacial bond between the CNTs and the AZ91 alloy.After Ni-CNTs were added to the AZ91 magnesium alloy, the interface reactant Mg_2_Ni was generated by in-situ reaction between the matrix and the surface of the CNTs. Mg_2_Ni plays a bridging role in the composite material, and the mechanical properties of the composite material are improved to some extent. The tensile strength and elongation of the 1.0 wt.% Ni-CNTs/AZ91 magnesium matrix composites were increased by 36% and 86% compared with the AZ91 matrix alloy, respectively.After Ni-CNTs were added to the AZ91 matrix alloy, more dimples were found at the fracture. The fracture behavior of the Ni-CNTs/AZ91 composite was transformed from cleavage fracture of the AZ91 matrix alloy to quasi-cleavage fracture. Meanwhile, CNTs dispersed near the fracture showed a “pull-out” state, which could effectively bear and transfer loads.


## Figures and Tables

**Figure 1 materials-15-07946-f001:**
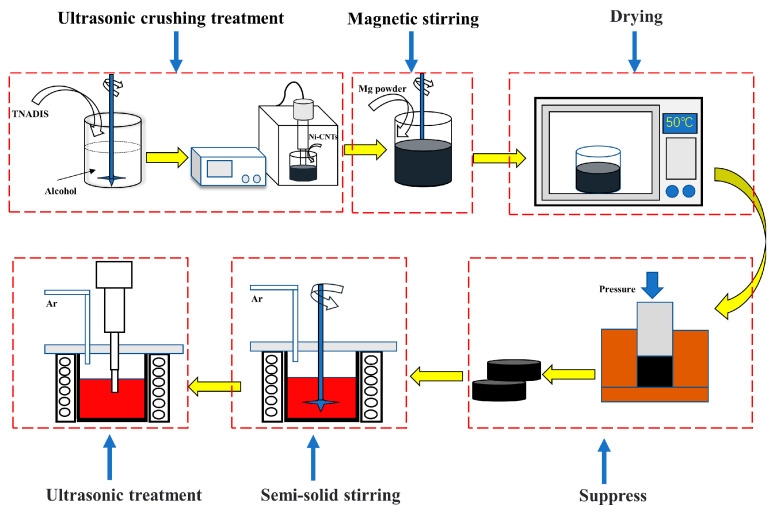
Schematic diagram of the preparation of Ni-CNTs/AZ91 magnesium matrix composites.

**Figure 2 materials-15-07946-f002:**
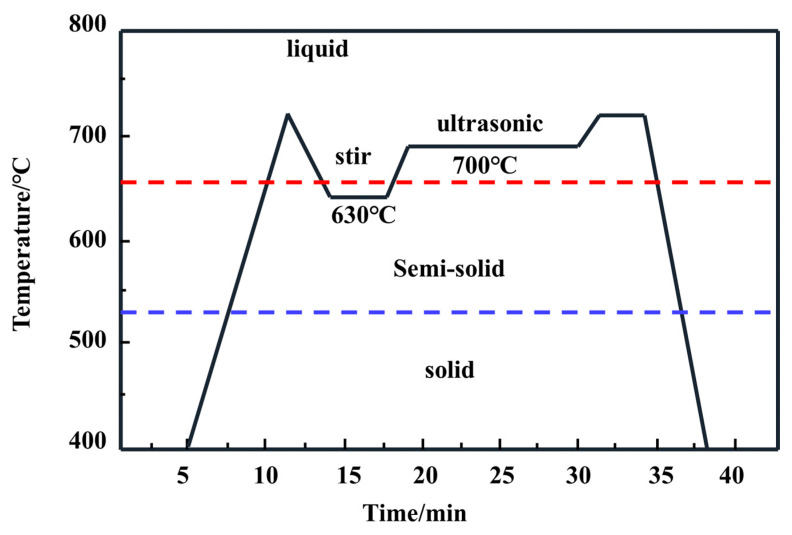
Flow chart of smelting process.

**Figure 3 materials-15-07946-f003:**
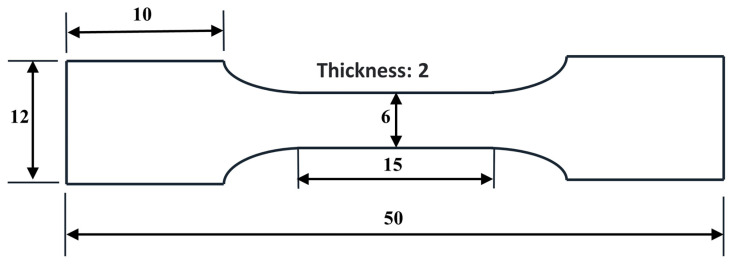
Tensile specimen size of Ni-CNTs/AZ91 magnesium matrix composites (unit: mm).

**Figure 4 materials-15-07946-f004:**
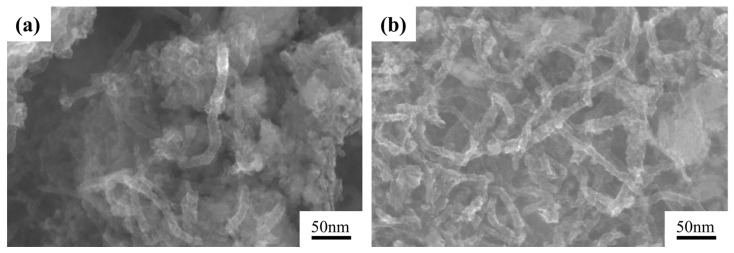
SEM image of surface morphology of Ni-CNTs (**a**) before and (**b**) after pre-dispersion.

**Figure 5 materials-15-07946-f005:**
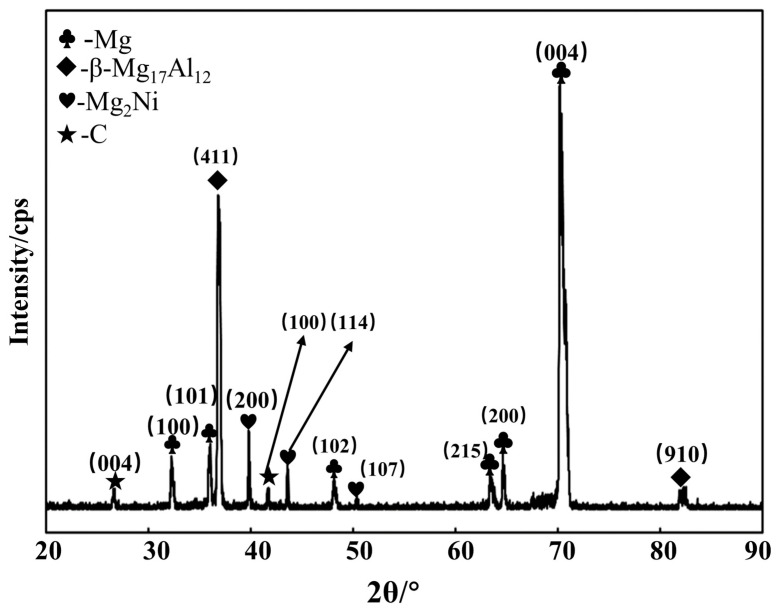
X-ray diffraction pattern of Ni-CNTs/AZ91 magnesium matrix composites.

**Figure 6 materials-15-07946-f006:**
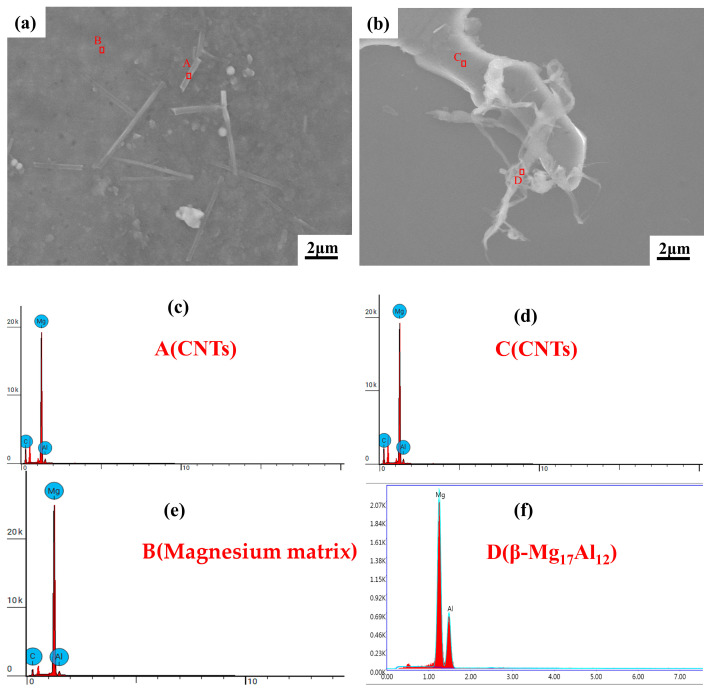
SEM images and EDS analysis of Ni-CNTs/AZ91 magnesium matrix composites: (**a**) uniformly distributed Ni-CNTs in the Mg alloy matrix (**b**) wound on the β-Mg17Al12 phase (**c**–**f**) EDS analysis: CNTs, Magnesium matrix, CNTs, β-Mg_17_Al_12_.

**Figure 7 materials-15-07946-f007:**
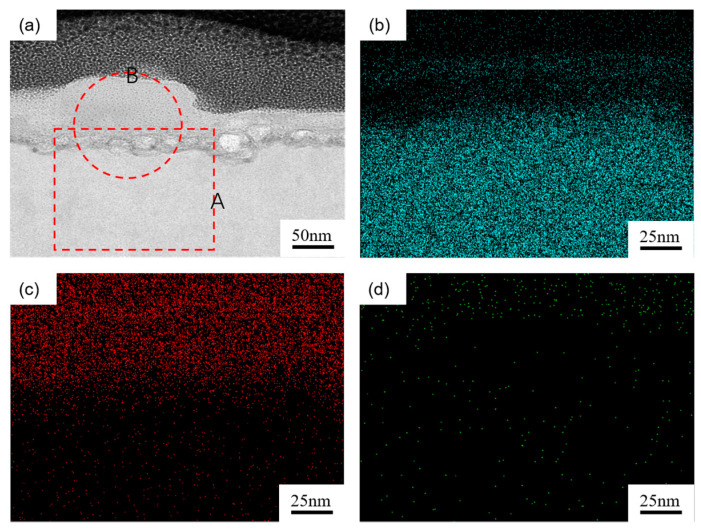
CNTs in Ni-CNTs/AZ91 composite materials, Mg and Mg_2_Ni distribution. (**a**): TEM micrographs; (**b**–**d**): EDS mapping of Mg, C, and Ni, respectively (region A).

**Figure 8 materials-15-07946-f008:**
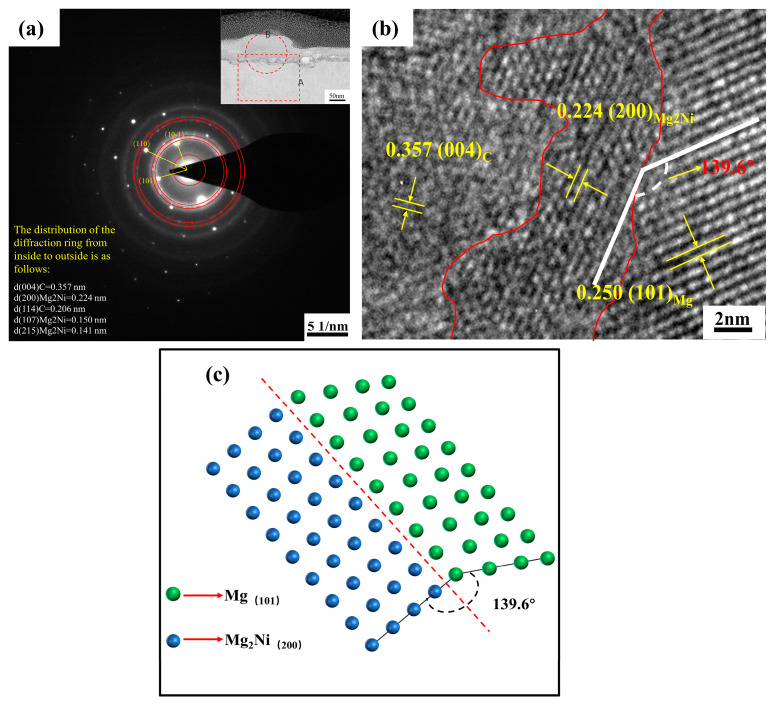
Electron diffraction pattern and interface structure of composite materials (**a**) electron diffraction pattern of region B; (**b**) Interface structure diagram of CNTs, Mg, and Mg_2_Ni (region A); (**c**) Schematic diagram of common lattice interface.

**Figure 9 materials-15-07946-f009:**
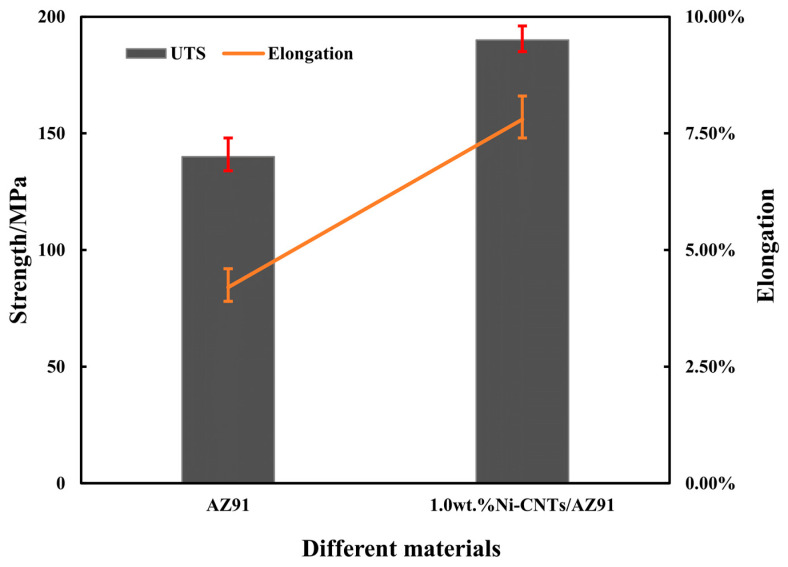
Tensile mechanical properties of the two materials (AZ91 matrix material, 1.0 wt.% Ni-CNTs/AZ91 magnesium matrix composite) at room temperature.

**Figure 10 materials-15-07946-f010:**
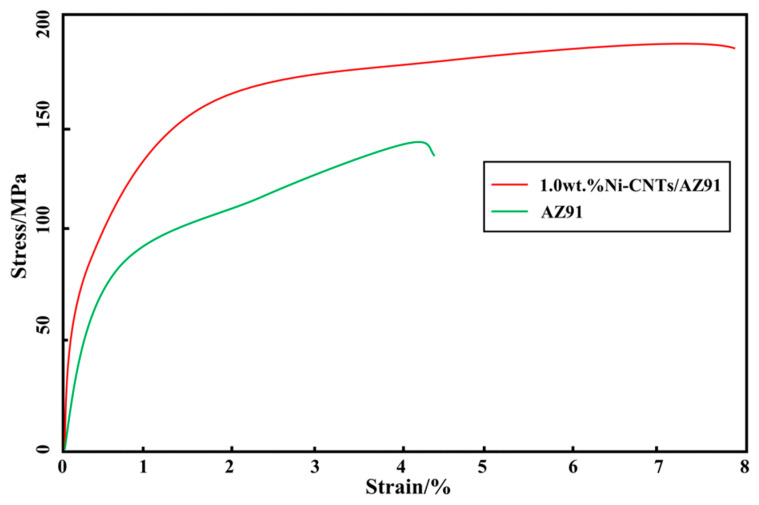
Tensile stress-strain curve of the two materials (AZ91 matrix material, 1.0 wt.% Ni-CNTs/AZ91 magnesium matrix composite) at room temperature.

**Figure 11 materials-15-07946-f011:**
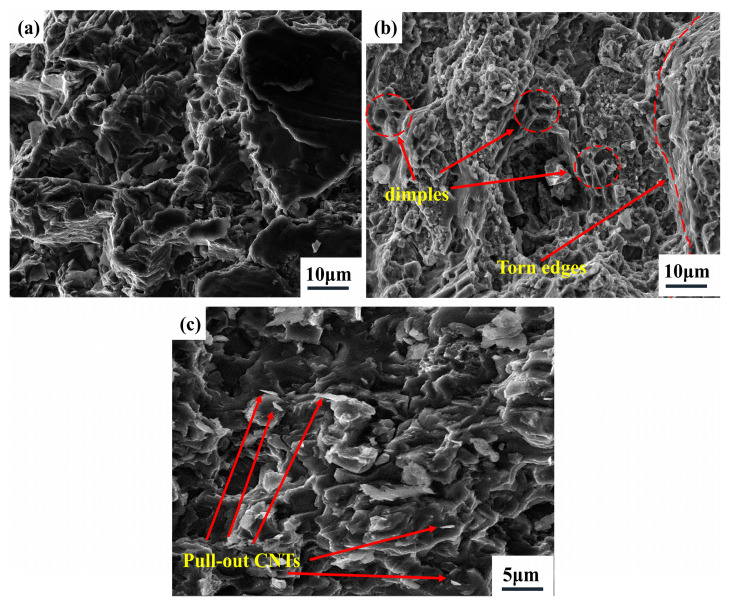
Fracture morphology of the two materials: (**a**) AZ91 matrix alloy; (**b**) and (**c**) 1.0 wt.% Ni-CNTs/AZ91 magnesium matrix composites.

**Figure 12 materials-15-07946-f012:**
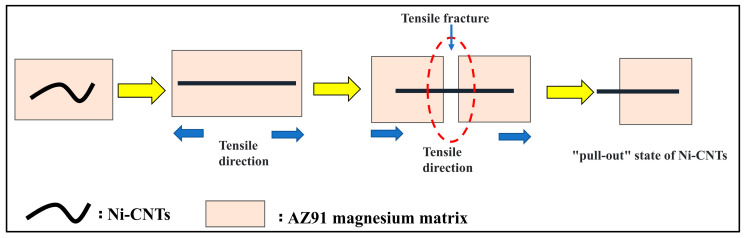
Schematic diagram of the tensile process of Ni-CNTs/AZ91 magnesium matrix composites.

**Table 1 materials-15-07946-t001:** Primary experimental materials.

Materials	Outer Diameter/nm	Particle Size/μm	CNTs Content	Manufacturer
AZ91 magnesium alloy	/	/	/	/
Ni-CNTs (nickel-plated carbon nanotubes)	20–30	10–30	>38%	Chengdu times nano
TNADIS dispersant	/	/	/	Chengdu times nano
Mg	/	100	/	Comio Chemical Reagent Factory

## Data Availability

Data generated or analyzed during this study are included in this published article.

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
