# Peer review of "Microstructure, Interface and Strengthening Mechanism of Ni-CNTs/AZ91 Magnesium Matrix Composites"

_materials, 2022, doi:10.3390/ma15227946_

Round 1

Reviewer 1 Report

The most important information of presented work was an increase of  mechanical properties by application of 1 wt.% Ni-MWCNTs instead of 1 wt% MWCNT in AZ31 magnesium alloy, and tensile tests curves for that illustration should be shown.

Unfortunately, in experimental procedure and results discussion I found a lot of  shortcomings  and faults.

-          There are no precious information and microstructure characterization both of Ni-MWCNTs and MWCNTs  applied for comparison. In article is “Fig.4 SEM image of surface morphology of CNTs (a) before and (b) after pre-dispersion” while in lines 108-109 “The morphology of Ni-CNTs obtained after pre-dispersion treatment is shown in Fig.4 (b)”.

-          Moreover,  the fundamental question is applying of the same  wt.% of MWCNTs and  Ni containing 30% of MWCNTs. What the Authors wanted  to compare in microstructure  and decohesion mechanisms?

-          Authors omitted  the distribution of Mg2Ni phase in microstructure characterization (visible clearly in Fig. 10c as very fine granular particles) and the role of intermetallics in strengthening of magnesium based materials.   In initial nanofibers the proportion of Ni:MWCNTs was like 2:1. Does it mean the presence of brittle Mg2Ni coating around MWCNTs or the intermetallic particles are dispersed in magnesium matrix only?

-          The scheme presented in Figure 11 does not show the real effects occurring in tensile test.

-          My general suggestion is to fabricate an additional reference material of the same MWCNTs volume fraction as in Ni-MWCNTs. That will allow  to compare the real role of MWCNTs in strength improving. Moreover, an application of  higher magnification  in materials microstructure and decohesion mechanisms analysis is necessary. For characterization of interface MWCNTs the TEM method with SADP is suggested – a presence of Mg2Ni in form of coating can be confirmed or excluded.

Reviewer 2 Report

Minor questions:

    Line 91: what was the X-ray source (what was the wavelength)?

    What was the source of SEM images? Secondary or backscattered electrons?

    Figure 9 and the corresponding results in lines 185-192: what are the margins of error of the measured data?

Major questions:

    Line 120 and figure 5.: what was the identified carbon phase? (eg. graphite)

    Lines 160-163 and figure 8(a): about "five kinds of carbon phases": these are not five carbon pahses. These are five diffraction spots / rings / crystal plane spacings etc. which correspond to 1-2 (or even more) carbon phases. What were these phases? How they are related to the (only one) carbon phase identified by XRD?

Major issues:

    Please explain more the reason of the topic selected, respecting that there were similar studies eg. AZ31 with CNTs: https://doi.org/10.1007/s12666-015-0702-x

    Lines 237-239: conclusion 2. is too technical; it lacks of the "because of...".

Typos, format remarks:

    Line 43: "MWCNTs"
    Line 68: "550 W"
    Line 78: "1 kW"
    Figures 5. and 9.: please be consistent: Intensity/cps, 2{theta}/°, Strength/MPa (or Intensity (cps), 2{theta} (°), Strength (MPa) )
    Line 132: a lowercase letter after a colon.
    Lines 160-167 and 185: "0.158 nm", "140 MPa" etc.
    Figure 7.: figure caption should be like: " CNTs in Ni-CNTs/AZ91 composite materials, Mg and Mg2Ni distribution. (a): TEM micrographs; (b)-(d): EDS mapping of Mg, C, and Ni respetively"
    Figure 8.: caption should mentions (better emphasize), that subfigures (a) and (b) correspond to regions marked on figure 7.
    It is a quite disturbing, that figure 8(a) is not the SAED of figure 8(b). Figure 8(a) should be an individual figure.
    Figure 8(b): white "139.6°" is very hard to read similar to the lower indices of lattice planes.
    Figure 10(c) (and (d)): it is very hard to read the red text.
    Line 292: "Kumar"

Reviewer 3 Report

Liu et al. have studied the microstructure, interface and strengthening mechanism of Ni-CNTs/AZ91 magnesium matrix composites. The below mentioned comments must be incorporated to improve the manuscript.

·         Why authors used CNTs as these are very expensive. Apart from this the reason also needs to discussed for the same.

·         Abstract and conclusions section needs to address some values regarding your main findings as well.

·         How many measurements were taken to determine the mechanical characteristics? The authors should provide values for mechanical characteristics taking into account experimental error.

·         Is the CNTs synthesized or purchased please include the details.

·         The novelty of the manuscripts needs to discussed carefully, there are many works have been available for the incorporation of the CNTs in AZ91. The simple google shows many similar results in other papers.

·         “Ni-CNTs was evenly distributed in the magnesium matrix” please add another SEM image which clearly justify this statement.  EDS mapping may be used.

·         “the whole experiment process was carried out under an argon atmosphere” the reason needs to be discussed.

·         Why authors not used different wt.% of the Ni-CNTs and CNTs in the matrix? Why others mechanical characteristics have been not presented?

·         “It can be seen that during the tensile process, the CNTs experienced successively from bending state to straight state” how do you confirm?

·         Figure 10: please use different colour for marking.

·         “Tearing edges and many dimples can be found in the fracture surface, the fracture surface is rough with large surface undulation”, I am not able to see please mark these points.

·         In the introduction, few literature are suggested to  be discussed 10.1007/s40962-021-00747-9, 10.3390/ma14040990

·         Grammar needs a revisit. The flow of language needs some work.

·         Some typos need to be corrected throughout the manuscript.

Reviewer 4 Report

The manuscript dealing with Ni-CNTs/AZ91 composites is focused on the study of the microstructure, interface and strengthening mechanism. The manuscript is well written and discussed. However, some major modifications need to be done before it can be recommended for publication. The main comments / questions are:

1. Although the purpose of experiments is clear, the manuscript lacks a motivation. It should be mentioned (explicitly, in Abstract and Introduction), why the work was done.

2. Why did the authors press the mix the Ni-CNTs/Mg powder into the blocks? From the homogeneity point of view, wouldn’t be better to add to the furnace (at 630°C) the powder mixture instead of pressed blocks? What are the dimensions of the blocks? What is the role of the Mg powder besides Mg in AZ91?

3. The microstructural analysis (chapter 3.1) is insufficient and should be discussed in more details. The SEM images lack the denotation of particular phases / particles. Moreover, the magnification used in Fig. 4a is too low to compare it to Fig. 4b. In Fig. 4a, it is hard to distinguish particular elements / phases / particles without any further information. An EDX map would show the distribution of Ni, Mg and C, which would help to the better description of the microstructure. What does indicate that Ni is in the form of a coating (line 122)?

4. In Fig. 5, phase denotation (Mg, C, Mg2Ni, β-Mg17Al12) coupled with Miller indices should be assigned to each individual peak. In that case, the solid symbols would be useless.

5. The authors state that a small number of winding distribution of CNTs on β-Mg17Al12 can be seen in Fig. 6b (lines 131-132). What is the experimental prove for the occurrence of β-Mg17Al12 in this image? The SEM image should also be coupled with EDX results to support this statement.

6. The scales (50nm) in the case of TEM-EDX elemental maps are questionable. If the particular maps (b-d) represent the A-region marked in Fig. 7a, the scales in (b-d) should be different from the a) image. Moreover, the maps seem no to correspond to the A-region. Please give a statement.

7. Since the “Mechanical properties” chapter deals with two types of composites (filled with CNTs or Ni-CNTs) besides the pure AZ91 alloy, the type of nanotubes should be unequivocally distinguished throughout the whole manuscript, otherwise it can be confusing. For example in Experimental procedure, the authors describe the preparation of the Ni-CNTs/AZ91 composite, however, they often use the simplified denotation CNTs. Please unify the denotation.

8. Quality / resolution of SEM images should be improved. Besides that I would suggest the following modifications to improve the visibility and readability of images in this manuscript:

- Fig. 1 – particular steps could be denoted with the process name

- Figs. 4 and 7 – to improve the informative value, the assignment of particular elements / phases / particles to the particular regions could be added into the image

- Fig. 6 – besides the resolution, both contrast and brightness should be tuned

- Fig. 10 – the description in (c-d) is hardly visible, please highlight it

9. The authors mention that at least three measurements were performed for each sample. Therefore, it can be concluded that the mean values are shown in Fig. 9. In that case, please include the standard deviations.

10. More detailed parameters of experimental techniques used in this work should be introduced in Experimental procedure, e.g. XRD – anode, measured 2Theta range, step size, time per step; SEM – acceleration voltage, imaging modes; TEM – acceleration voltage; tensile test – testing speed, etc.

The manuscript also needs some minor / formal modifications:

- line 19: were ---> was

- line 37: to improving ---> to improve

- line 39: its’ ---> its

- lines 39-40, the sentence: “CNT was first discovered by Iijima in 1991 [16, 17], which has excellent mechanical properties…”, better: “CNT, discovered by Iijima in 1991 [16, 17], has excellent mechanical properties…”

- line 43: MCWNTs ---> MWCNTs

- line 63: were ---> are

- lines 69-70: CNTs-Mg powder mixed powder

- lines 98 and 158: was shown ---> is shown

Round 2

Reviewer 1 Report

Text should  be very carefully check by language edutor.

I found many error and inaccurances.

l.50-51 In order to 50 solve the wettability of CNTs and Mg, Han et al. [26] prepared Ni-CNTs/Mg matrix composite material by hot extruded

The effective interface bonding between CNTs and Mg matrix, but the dispersion of CNTs in Mg matrix was poor, and the ideal strengthening  effect on Mg matrix is not achieved (meaning of the sentence?))

EXAMPLES

l.50-51 In order to 50 solve the wettability of CNTs and Mg, Han et al. [26] prepared Ni-CNTs/Mg matrix composite material by hot extruded

The effective interface bonding between CNTs and Mg matrix, but the dispersion of CNTs in Mg matrix was poor, and the ideal strengthening  effect on Mg matrix is not achieved (meaning of the sentence?))

The agglomeration of Ni-CNTs and Ni-CNTs in AZ91 magnesium matrix can be solved by ultrasonic broken and melting ultrasonic (meaning of the sentence?)

Reviewer 3 Report

The authors have addressed al the comments. It may be accepted for publication.

Author Response

Thank you for your comments on this article.

Reviewer 4 Report

Authors sufficiently responded to all comments and questions. Before the publication, I have the following minor suggestions:

- to make the manuscript consistent, please include TEM parameters to Experimental materials and methods (similarly to SEM)

- Fig. 6 - added EDX spectra are difficult to read. Please, try to make them more visible.

- line 52: hot extruded ---> hot extrusion
